# The Main Changes in Pregnancy—Therapeutic Approach to Musculoskeletal Pain

**DOI:** 10.3390/medicina58081115

**Published:** 2022-08-17

**Authors:** Felicia Fiat, Petru Eugen Merghes, Alexandra Denisa Scurtu, Bogdan Almajan Guta, Cristina Adriana Dehelean, Narcis Varan, Elena Bernad

**Affiliations:** 1Department of Obstetrics-Gynecology II, Faculty of Medicine, “Victor Babes” University of Medicine and Pharmacy, Eftimie Murgu Square, No. 2, 300041 Timisoara, Romania; 2Department of Physical Education and Sport, Banat’s University of Agricultural Sciences and Veterinary Medicine “King Mihai I of Romania” from Timisoara, Calea Aradului 119, 300645 Timisoara, Romania; 3Department of Toxicology and Drug Industry, Faculty of Pharmacy, “Victor Babes” University of Medicine and Pharmacy, Eftimie Murgu Square, No. 2, 300041 Timisoara, Romania; 4Research Centre for Pharmaco-Toxicological Evaluation, “Victor Babes” University of Medicine and Pharmacy, Eftimie Murgu Square, No. 2, 300041 Timisoara, Romania; 5Department of Physical Therapy and Special Motor Skills, Faculty of Physical Education and Sport, West University of Timisoara, Vasile Parvan Boulevard, No. 4, 300223 Timisoara, Romania

**Keywords:** pregnancy, changes of pregnancy, musculoskeletal pain, treatment

## Abstract

*Background and Objectives*: During pregnancy, women undergo various physiological and anatomical changes that are accentuated as the pregnancy progresses, but return to their previous state a few weeks/months after the pregnancy. However, a targeted therapeutic approach is needed. Most of the time, during this period, these changes precipitate the appearance of pain, musculoskeletal pain being the most common. Pregnant women should avoid treating musculoskeletal pain with medication and should choose alternative and complementary methods. Exercise along with rest is the basis for treating chronic musculoskeletal pain. Side effects of physical therapy are rare and, in addition, it is not contraindicated in pregnant women. The benefits of this type of treatment in combating pain far outweigh the risks, being an easy way to improve quality of life. The objective of this article is to discuss the management of musculoskeletal pain during pregnancy, to identify the main musculoskeletal pain encountered in pregnant women along with drug treatment, and to expose the beneficial effects of alternative and complementary methods in combating pain. *Materials and Methods:* A literature search was conducted using medical databases, including PubMed, Google Scholar, and ScienceDirect, using the keywords “changes of pregnancy”, “musculoskeletal pain”, “pregnancy pain”, “pain management”, “pharmacological approach”, “alternative and complementary treatment” and specific sites. Information was collected from studies whose target population included pregnant women who complained of musculoskeletal pain during the 9 months of pregnancy; pregnant women with other pathologies that could increase their pain were not included in this review. *Results*: The articles related to the most common non-obstetric musculoskeletal pain in pregnancy along with pharmacological treatment options and alternative and complementary methods for musculoskeletal pain management during pregnancy were selected. *Conclusions*: The results were used to guide information towards the safest methods of therapy but also to raise awareness of the treatment criteria in order to compare the effectiveness of existing methods. Treatment must consider the implications for the mother and fetus, optimizing non-pharmacological therapeutic options.

## 1. Introduction

During pregnancy, a woman’s body undergoes various changes, both physiological and anatomical, necessary to meet the increased metabolic needs in order to support the growing fetus, for its harmonious development and also to prepare the body for birth [1,2]. It has been observed that the first changes appear in the first trimester of pregnancy and intensify once the final term is reached and return to normal a few weeks after birth [2]. The physiological changes produced by pregnancy are generally well tolerated by healthy women, but certain changes are still likely to aggravate different pathologies or give rise to a variety of disorders, especially musculoskeletal, which is the most common in pregnancy [3]. A woman’s body undergoes major changes during pregnancy in all organs to support both the mother and the fetus.

A proper diet is important at any time of life, especially during pregnancy. The diet of the pregnant woman must provide nutrients and energy to support both the normal requirements of the mother and the needs of the developing fetus. The recommended diet for pregnant women does not differ much from the diet of an adult, necessarily comprising a healthy, balanced and varied diet. For pregnant women, it is recommended to use foods rich in minerals and vitamins, especially iron, vitamin D and folate [4]. Additionally, at the time of conception, a balanced diet is required, and it is preferable for the woman to have a healthy body weight with a body mass index of 20–25, since lower or higher values can influence the fertility and harmonious growth of the fetus [5]. Excessive weight gain during pregnancy has countless negative effects on the health of the mother and child, including high blood pressure, diabetes, birth trauma and asphyxia. This weight gain in pregnant women is associated with a higher risk of long-term obesity but also with the precipitation of musculoskeletal pain [6]. Moreover, aside from diet, the evolution of pregnancy is influenced by a multitude of factors, including age, previous experiences and family history, as well as stress, smoking and excessive alcohol or coffee consumption [7]. Both extremes of age considered appropriate for the conception of the child can cause unwanted effects on pregnancy. Young mothers have a high risk of low birth weight, premature birth and fetal death resulting from biological immaturity, lack of access to prenatal care to socioeconomic factors. However, the mother’s advanced age is more strongly associated with complications at birth [8]. There is a close relationship between healthy mothers and healthy newborns, and proper nutrition and physical activity are key factors that contribute to its achievement; factors that can prevent and combat musculoskeletal pain encountered during pregnancy. Special attention was paid to physiotherapy in pregnancy with the help of the American College of Obstetrics and Gynecologists guidelines in 2002 [9], which led to studies focusing on the beneficial effects of exercise during pregnancy.

New detailed clinical trials are currently needed regarding the treatment of the most common pain in pregnancy, adapted to current times with socio-economic or dietary modifications; a treatment that must be correlated with physiological and anatomical changes that normally occur in pregnant women. In the foreground should be non-pharmacological treatment; physiotherapy being a method more easily accepted by pregnant women. This paper provides a comprehensive look at the physiological and anatomical changes in pregnancy, the main musculoskeletal problems and the treatment that can be addressed, pharmacological treatment and especially non-pharmacological treatment, based on alternative and complementary methods.

## 2. Physiological and Anatomical Changes of Pregnancy

### 2.1. The Main Changes That Occur at the Organ Level

Regarding the cardiovascular system, during pregnancy heart rate can increase by up to 60% with the highest values registered from week 20 until birth. Cardiac output is required to increase and to maintain normal blood pressure. Initially, the increase in cardiac output is achieved by an increase in stroke volume, followed at the end of the third trimester by an increase in heart rhythm [2,10]. The highest increase in cardiac output is achieved in the kidneys, uterus and skin to control the mother’s temperature and to produce nutrients needed for the fetus and to eliminate fetal and maternal waste [11,12,13]. The presence of the pregnant uterus leads to the lateral movement of the heart. In the first weeks of pregnancy, the pregnant uterus begins to cause the mechanical compression of the inferior vena cava and the descending aorta. Thus, a reduction in cardiac output and venous return is observed, resulting in maternal hypotension and fetal acidemia. To correct and compensate for compression, heart rate and sympathetic tone increase. In many pregnant women, these mechanisms may be insufficient to support blood pressure and aortocaval compression syndrome may set in. This hypotensive syndrome is accompanied by symptoms such as pallor, dizziness, sweating and tachycardia, followed by bradycardia and hypotension in the supine position, while the severe form can cause death [14,15].

During pregnancy, in the gastrointestinal tract, gastric motility is most affected. High levels of hormones, especially progesterone, cause smooth muscle relaxation and decreased bowel motility, thus prolonging gastric emptying time, leading to constipation [16]. Likewise, the progesterone-mediated relaxation of the lower esophageal sphincter induces a decrease in its tone manifested as gastroesophageal reflux disease [17]. Additionally, high hormone levels cause vomiting and nausea, known as morning sickness, which occur at any time of the day in more than 70% of pregnant women. However, if these conditions increase after week 20 or lead to ketosis with a massive weight loss, hyperemesis gravidarum can be reached and intravenous vitamins and fluids may be needed [18].

Respiratory changes also occur in pregnant women. The diaphragm rises by about 2 cm, which leads to a 5% decrease in lung capacity. The current volume increases by up to 40% leading to a decrease in the expiratory reserve volume by 20% and an increase in ventilation per minute (VM). The increase in VM determines an increase in the level of arterial (PaO_2_) and alveolar (PAO_2_) pressure but also a decrease in the partial pressure of arterial carbon dioxide (PaCO_2_). The decrease in PaCO_2_ produces an increased gradient of carbon dioxide (CO_2_) between mother and fetus, helping to deliver oxygen to the fetus and eliminate CO_2_. Elevated progesterone levels create this gradient, with progesterone being a respiratory stimulant that sensitizes CO_2_ receptors [19,20,21]. Dyspnea in pregnancy, which occurs in about 70% of pregnant women is due to these low levels of PaCO_2_ as well as decreased total lung capacity and increased VM. Thus, most often in the third trimester, the pregnant woman may experience a feeling of shortness of breath [1].

In addition, renal changes were observed; renal plasma flow and glomerular filtration rate were increased. This increase in glomerular filtration rate maintains plasma sodium levels, which increase because of the activation the renin–angiotensin–aldosterone system, and also lowers serum creatine and urea nitrogen in the blood [20]. At the renal level, elevated progesterone levels work to reduce peristalsis, urethral tone and contraction pressure, producing renal vasodilation. Furthermore, hormone levels, but also external mechanical compression and changes in the urethral wall, can cause hydronephrosis and hydroureter in the pregnant woman. Urinary tract infections, urinary incontinence and nocturia are common in pregnancy. All changes that occur during this period return to the previous state up to 6 weeks after birth [13].

Additionally, during pregnancy, plasma volume increases by up to 50% and erythrocyte volume by up to 30%. These significant changes can lead to physiological anemia and decreased hematocrit, an effect that reduces blood viscosity and resistance to blood flow. Regarding the number of leukocytes, they increase and can reach very high values during labor, which can often explain the severity of the infection; however, these values do not make the pregnant woman more prone to infections [22]. In general, the concentration of platelets is maintained at normal values, but there are some cases in which their number decreases (gestational thrombocytopenia) due to the increase in plasma volume, which disappears after birth [23]. Fibrinolytic and coagulation pathways also undergo changes, and venous stasis leads to an increased risk of thromboembolism, beginning in the first trimester of pregnancy and lasting up to 3 months after birth [20].

In pregnant woman, high levels of estrogen stimulate the production of the thyroid-binding globulin, leading to an increase in the total levels of triiodothyronine (T3) and thyroxine (T4) by about 50%, while free T3 and T4 levels remain constant or slightly altered. It also increases the production of hormones by the adrenal glands. Low blood pressure and vascular resistance stimulate the renin–angiotensin–aldosterone system, resulting in an up to 10-fold increase in aldosterone at the end of the third trimester of pregnancy. Moreover, there is an increase in the level of cortisol and adrenocorticotropic hormone, a corticosteroid-binding globulin, which leads to a state of hypercortisol. On the other hand, the high levels of estradiol in pregnancy result in an increase in prolactin that induces growth in the pituitary gland [1,2,20].

### 2.2. Musculoskeletal Changes

Over time, it has been observed that many physiological and anatomical changes occur in pregnant women that can affect the essential organs, but the most common are those at the musculoskeletal level. Weight gain, the enlargement of the uterus with a shift in the center of gravity and hormonal and vascular changes cause a number of musculoskeletal problems. Changing the center of gravity causes lumbar lordosis with the flexion of the neck and the drooping of the shoulders [24,25].

Mechanical pressure, elevated progesterone and relaxin levels increase joint laxity and prepare a woman’s body for childbirth. Additionally, in pregnant women, fluid retention causes the compression of soft tissues. All these changes that occur make the pregnant woman susceptible to musculoskeletal disorders. Most pregnant women complain of musculoskeletal disorders, and some of them show signs of disability. The majority conditions have been described as spinal pain, pain in the upper and lower extremities, peripheral neuropathy and muscle cramps. In addition, carpal tunnel syndrome, which results from compression of the median nerve, is quite common in pregnant women [19,20,26]. Major changes that occur in the body of pregnant women at different system level are shown in Figure 1.

These musculoskeletal disorders cause disabilities and loss of work capacity in pregnant women [27]. Pain during pregnancy can be of various causes and is not limited only to labor pain. The non-obstetric causes of pain in pregnancy are common, from acute conditions, such as infection or injury, to secondary pain from underlying medical conditions. Pregnancy manifests physiological effects on a woman’s body, likewise influencing the endocrine, cardiovascular and renal systems as well as the musculoskeletal system. Additionally, weight gain during pregnancy and the presence of the pregnant uterus put pressure on the skeletal system. During the nine months of pregnancy, various changes occur in the body. These physiological and anatomical changes precipitate the onset of pain or may exacerbate pre-existing painful disorders.

Next, some aspects of the main imaging investigations for the diagnosis of musculoskeletal disorders will be presented.

X-rays are among the oldest types of imaging available, which use electromagnetic waves to produce an image. Bone lesions can be highlighted by radiography, but in order to diagnose muscle disorders more advanced imaging explorations are needed. The possibility that an X-ray taken during pregnancy will produce serious effects on the fetus is very small. However, the risk of affecting the child depends on the amount of radiation exposure and its gestational age. Thus, the exposure to high doses of radiation in the first two weeks after conception can cause a miscarriage; exposure up to eight weeks can increase the risk of congenital malformations; and up to week 16, high doses of radiation can lead to intellectual disability. The dose of a single exposure to radiation is far lesser than the doses associated with these risks. Another method of diagnosing musculoskeletal disorders is computed tomography (CT), which investigates bone and muscle injuries. This method offers a more detailed examination of body compared to radiography. It is not proven that the fetus is affected by the amount of radiation used in CT. However, its use on the abdomen or pelvic area is to be avoided in order to exclude any risk of developing cancer in childhood. If the examination of both the mother and the fetus is necessary, other imaging tests, such as ultrasound or magnetic resonance imaging (MRI), are used. Ultrasound uses sound waves to photograph the inside of the body. Ultrasound captures images of soft tissues, muscles and ligaments and can also diagnose musculoskeletal conditions along with MRI. Ultrasound can easily detect the signs of inflammation at the muscle and joint level. This imaging method is the most used for pregnant women, but if the images obtained do not provide a clear answer, an MRI examination can be resorted to. MRI is a method that uses radio waves and magnetic fields to capture images of the inside of the body. MRI can capture images of the body’s soft tissues, even muscles, unlike X-ray imaging. In addition, MRI can capture joint injuries, such as torn ligaments or cartilage. There are no proven adverse effects due to the MRI either in the pregnant woman nor in the fetus, and is a frequently used method today [28,29].

The main musculoskeletal pains encountered during pregnancy can be classified into lumbar, pelvic and joint pain.

#### 2.2.1. Lower Back Pain

Lower back pain occurs in about half of pregnant women and is considered a normal pain during pregnancy [30]. A higher risk of lower back pain is associated with age, the presence of pain before pregnancy and especially during menstruation, and ethnicity, particularly in African American and Caucasian individuals; although individuals of Hispanic ethnicity do not show strong associations with lower back pain during pregnancy, nor do factors such as caffeine, tobacco, oral contraceptive medication, parity and exercise [31,32]. Additionally, other factors that can contribute to lower back pain are the mechanical compressions of the pregnant uterus that modifies the center of gravity and increases the force applied to the spine, as well as the pelvic ligament laxity and vascular compression.

In addition, lower back pain encountered at night during pregnancy may be due to venous engorgement in the pelvis. The growing uterus presses on the vena cava and combined with fluid retention in pregnancy causes venous congestion and hypoxia in the lumbar spine [33]. Some women experience this pain from the first trimester when the mechanical pressure is not high, in this case the hormonal level being the factor that influences the appearance of the musculoskeletal disorder [31]. Most women recover easily in the first months after birth, of which only 50% seek professional medical assistance. The rapid identification of pain and the application of specific methods of treatment according to the particularities of each individual ultimately leads to total recovery. However, pregnant women who have gained more weight have a higher risk of postpartum lower back pain [34].

The most affected musculoskeletal structures during pregnancy and postpartum are the pubic symphysis and the sacroiliac joints. Most of the time it is difficult to establish the exact cause of the pain based only on the anamnesis, and detailed imaging investigations are required. Non-inflammatory causes of back pain, such as mechanical stress on the pelvic area caused by pregnancy, can cause subchondral bone marrow edema that cannot be differentiated from axial spondyloarthritis [35].

Bone marrow edema (BME) is a condition encountered in radiology described as a nonspecific lesion pattern, characterized by a change in marrow signal intensity at the level of the femoral head suggesting marrow infiltration through interstitial edema. Many aspects can lead to bone marrow changes, especially at the level of the sacroiliac joints, it is not yet clear whether postpartum bone marrow changes, which are mechanically induced, are different from inflammatory sacroiliitis seen on MRI. From an imaging point of view, spondyloarthropathy is characterized by BME around the sacroiliac joint (sacroiliitis) and structural changes, such as the fatty replacement of the bone marrow at the sites of inflammation, subchondral erosions and sclerosis [36].

The scientific team led by Agten compared the MRI results of the sacroiliac joints subjected to mechanical stress due to pregnancy with those of a group of women known to have spondyloarthritis to make the differential diagnosis. It was found that mechanical pressure on the joints can lead to bone marrow edema, only inflammatory changes being present, while in the group with spondyloarthritis, structural changes and erosions were also evident. Thus, these findings may provide guidance to differentiate mechanically- and hormonally-induced bone marrow changes and inflammatory sacroiliitis [37].

#### 2.2.2. Pelvic Pain

Pelvic pain is described by pregnant women as a burning sensation, stabbing in the sacral area or as pain in the pubic symphysis. This pain may radiate to the groin or posterior thigh. The first symptoms appear during week 18 and reach a maximum intensity in week 36 [38]. Pelvic girdle pains are not pains that normally occur in pregnancy, so it is necessary to intervene as soon as possible in their treatment, otherwise they can lead to severe pain. If timely therapeutic measures are taken, recovery is rapid. Up to 22% of pregnant women may suffer from pelvic pain and up to 8% may experience severe symptoms [39]. The causes of pelvic pain in pregnancy are multifactorial. Increased movement of the pelvic girdle causes pain due to increased ligament laxity caused by high levels of relaxin and estrogen. These high concentrations of hormones lead to enlargement of the pubic symphysis, which results in pain due to the increased mobility of the joints. Pelvic pain is amplified by mechanical exertion, anterior back pain or anterior pelvic trauma [26,40].

Furthermore, it has been shown that high body mass index, multiparity, mental stress, physical exhaustion and smoking are factors that can accelerate the onset of pelvic pain in pregnancy [41].

Lumbar disc herniation is considered the most common pathology of the spine in pregnant women; however, the condition is very rare in pregnant women compared to pelvic pain, with an estimated occurrence of 1 in 10,000 women. In recent years, an increase in the average age of women who become pregnant has been observed, a consideration that may increase the incidence of lumbar disc herniation among pregnant women. The most frequent symptoms were represented by radicular pain, along with the weakness of the muscles innervated by the root of a spinal nerve, the reduced sensation in the sensory distribution of a spinal nerve but also urinary incontinence. According to data from the literature, no more than 15% of patients suffering from this disorder develop severe neurological deficits. In terms of therapeutic management, it has been observed that the majority of lumbar disc herniation sufferers do not require surgery [42].

#### 2.2.3. Joint Pain

Pregnant women may experience joint pain, which often raises the suspicion of inflammatory diseases, such as rheumatoid arthritis or systemic lupus erythematosus. However, the development of new-onset inflammatory arthritis is rare during pregnancy, and there is evidence that pregnancy protects against new-onset rheumatoid arthritis [43]. Soft tissue swelling as well as joint laxity are physiological changes considered predisposing factors for the development of joint pain. In addition, hormonal changes with increased levels of progesterone, estrogen, relaxin and cortisol are associated with joint symptoms, stiffness and even arthralgia [44]. A study by Choi et al. aimed to establish the incidence of arthritis and arthralgia among pregnant women. Thus, of the 155 healthy pregnant women in the study, 9% developed arthritis and 16% had arthralgia. These conditions intensified in the third trimester of pregnancy, but the prognosis was generally good, with most of the disorders improving rapidly, concluding that the proximal interphalangeal joint of the hand was the most affected [45].

Another common condition is carpal tunnel syndrome, which occurs in the third trimester of pregnancy and affects the wrist. Due to water retention during pregnancy, the median nerve is subject to high compression. The symptoms are characterized by pain, paresthesia and tingling in the distribution area of the median nerve, which intensify after repeated movements and during the night [46]. Carpal tunnel syndrome developed in pregnancy has a benign course, most symptoms disappear after birth. However, a study of 45 pregnant women found that 49% still had symptoms 3 years postpartum [47]. Pregnant women undergo changes in every part of the body, and Figure 2 presents the most representative changes in terms of posture.

## 3. Treatment of Musculoskeletal Pain with Medication

The treatment of musculoskeletal pain should include a structured approach, combining non-pharmacological and pharmacological options, while surgery is only desired in extreme situations, such as for acute disc herniation [48]. For the treatment of pregnant women, the basic principle that must be observed is to reduce the use of drug therapy as much as possible, taking into account that it can have harmful effects on the fetus, the mother and the harmonious development of pregnancy. Many aspects need to be considered when setting up a treatment for a pregnant woman. A major limitation in the administration of drugs in pregnancy is the presence of potential embryotoxic and teratogenic effects of the therapeutic agent.

Liposolubility, molecular weight, maternal metabolism speed and the binding to plasma proteins are the main parameters on which the presence of drugs in the fetal and placental circulation depends. Most therapeutic agents reach the placental circulation to a greater or lesser extent, with the exception of large molecules, such as insulin and heparin. The stage with the highest teratogenic risk is the one between weeks 4 and 10 during the development of the organs. After this period, the action of drugs that cross the placental barrier can reduce the amount of amniotic fluid, can delay the birth of the fetus or can trigger certain pathologies [49]. A proportion of 3% of newborns have an obvious congenital malformation at birth, of which only 25% develop genetic malformations, the rest representing different causes, including the excess of drugs in pregnancy. On the other hand, an impediment in the investigation of the teratogenic effect of therapeutic agents is the specificity of the species. A well-known example is the administration of thalidomide, which caused severe malformations of the limbs of newborns, although studies on non-primates have not shown a teratogenic effect [50].

In the late 1970s, the Food and Drug Administration (FDA) implemented a cataloging system that assesses the potential risks of pregnancy-administered drugs. This system, based on scientific evidence, has classified drugs into five broad categories (Table 1). However, despite knowledge of these aspects, the risks of using pain medication during pregnancy are incomplete, and the benefit/risk balance must be analyzed in detail before initiating drug treatment in the pregnant woman [51].

### 3.1. Non-Opioid Medications

#### 3.1.1. Paracetamol (Acetaminophen)

Paracetamol is the most widely used analgesic in pregnancy. It has an analgesic potency similar to aspirin, being considered an effective and safe therapeutic agent at standard therapeutic doses throughout pregnancy. Acetaminophen is the main active substance used by pregnant women to combat mild to moderate pain and fever, due to its painkiller and antipyretic effects [68]. There are no studies showing congenital side effects. A study of thousands of pregnant women has shown the safety profile of acetaminophen, with no risk of malformations or other adverse effects of pregnancy [69,70]. Paracetamol has been reported to possibly present a high risk of attention deficit hyperactivity disorder when administered for more than 6 weeks, but according to the FDA these data are inconclusive [71]. Finally, several studies have evaluated the relationship between acetaminophen exposure in pregnancy and abnormalities. Various observational studies have investigated the effect of paracetamol on the reproductive and urogenital systems in thousands of mother–child pairs. Some of these studies have suggested that prenatal exposure to acetaminophen poses a high risk of cryptorchidism [72,73] and short anogenital distance (AGD), which is an indicator of genital masculinity [74,75]. In addition, another study found a possible link between paracetamol exposure and early puberty in females [76]. However, other studies have shown that there is no major risk of hypospadias after exposure to paracetamol in pregnancy [77,78,79].

Moreover, regarding the effects observed at the genital level following prenatal paracetamol exposure, possible neurological side effects have been identified. Observational studies have reported several associations between the occurrence of neurological disorders and the administration of acetaminophen during pregnancy, disorders such as autism [80,81,82], attention deficit hyperactivity disorder, decreased IQ (intelligence quotient) and speech/language delays, with exposure time having a significant influence [83,84,85,86,87,88,89,90]. Although paracetamol is considered a safe medication, based on the information gathered on the implications for the pregnant woman, it is recommended to be administrated with caution at the lowest effective dose and for the shortest period of time. These possible anomalies are summarized in Figure 3.

#### 3.1.2. Non-Steroidal Anti-Inflammatory Agents

The activity of NSAIDs (non-steroidal anti-inflammatory drugs) on the fetus is mediated by inhibiting the production of prostaglandins, which stimulate blood circulation in the organs. The effects of nonsteroidal anti-inflammatory drugs on the fetus differ depending on the period of pregnancy in which the mother used them [91]. In general, nonsteroidal anti-inflammatory agents are divided into groups according to their chemical structure and selectivity as follows: derivatives of salicylic acid (aspirin), propionic acid (naproxen, ibuprofen), acetic acid (diclofenac, indomethacin), enolic acid (piroxicam, metamizole) and selective cyclooxygenase-2 (COX-2) inhibitors (celecoxib). Figure 4 shows the chemical structures of the most used substances from the NSAID class by pregnant women. NSAIDs should be avoided during pregnancy, in the first trimester there is a risk of miscarriage [92,93] and in the third trimester, these drugs may have a negative effect on fetal circulation due to premature closure of the arterial canal and may present a risk of oligohydramnios [94]. Treatment with NSAIDs can cause reversible infertility in humans, as the inhibition of cyclooxygenases prevents normal reproductive processes. This therapeutic class is frequently used by young women for the treatment of various pathologies, even in the treatment of unruptured luteinized ovarian follicles. After stopping the administration of NSAIDs, ovulation returns to normal. However, this effect can be used in the planning of in vitro fertilization treatments. A representative that has demonstrated this action in several studies is indomethacin. Indomethacin has been shown to prevent follicular rupture and reduce premature ovulation, showing an increase in the recovery rate of immature eggs [95]. In addition, the administration of NSAIDs in the third trimester of pregnancy can narrow the arterial channel by inhibiting COX-2 and can cause pulmonary hypertension in the newborn. Diclofenac is the representative of the class that shows the greatest power to inhibit prostaglandin E2 (PGE2) production both in vitro and in vivo. Diclofenac is a low molecular weight molecule that easily crosses membranes, with an average maternal/fetus drug ratio < 1, concluding that over time it can accumulate in the fetal tissue [96,97]. Metamizole sodium, withdrawn from the US but still available in Europe and Latin America, has been studied for its toxic effect on the fetus. In Brazil, studies thereof have not been associated with birth defects, intrauterine death or premature birth [98,99,100]. On the other hand, there are studies that associate the administration of dipyrone with the appearance of Wilms tumors in children and in addition associate this drug with general class side effects, such as oligohydramnios, by decreasing renal perfusion in the fetus [101]. Likewise, NSAIDs are drugs associated with increased labor duration and onset of labor. Many pregnant women are exposed to this class of drugs, such as ibuprofen or naproxen, because they are released without a prescription. Aspirin, at doses up to 80 mg/day, has not been shown to pose a major risk to the mother and fetus, but when given in high doses, it can cause intracranial hemorrhage in premature babies born before week 35 [102]. The administration of low-dose aspirin (LDA) between 80 and 150 mg/day is currently one of the key interventions for the prevention of preeclampsia, along with optimal calcium intake and an appropriate lifestyle. Preeclampsia can cause severe complications for both the mother and the fetus. Most guidelines recommend LDA as prophylaxis for women at high risk of preeclampsia, but the method of identifying women at this risk has not been clearly established [103].

### 3.2. Opioid Medications

According to the FDA classification, codeine and most opioid compounds are in category C with potentially toxic to the fetus. The main compounds in this class are morphine-like agonists (hydromorphone, hydrocodone, codeine, oxycodone), meperidine-like agonists, and synthetic opioid compounds (tramadol). A study of 563 pregnant women showed that mothers’ exposure to codeine (Figure 5a) in the first trimester could lead to the birth of children with respiratory problems, with eight cases reported in this study [50].

In the case of pregnant women undergoing chronic opioid treatment, it is recommended that the lowest dose be used or, if possible, that the therapeutic agent be discontinued gradually, initially being replaced with the methadone of buprenorphine, to prevent withdrawal symptoms [104]. It has been observed that buprenorphine is an opioid medicine, which has some advantages, namely that it rarely causes withdrawal symptoms after stopping the administration and, in addition, mothers who receive treatment with this substance give birth to children with higher weight and who need shorter hospitalization times compared to other opioids [105]. Meperidine is not recommended for repeated use due to its metabolite, normeperidine, which accumulates in the body and produces exciting effects on the CNS (Central Nervous System). Whenever it is necessary to resort to opioid treatment, it is recommended to administer opioids without active metabolites [106]. Fentanyl (Figure 5b) patches are recommended as a treatment during pregnancy and even during breastfeeding; however, in this case also, great attention must be paid to the withdrawal symptoms [107].

### 3.3. Transdermal Therapies

Capsaicin and lidocaine in the form of topical treatment can be used to treat lower back pain. Topical lidocaine 5% inhibits fast sodium channels in neuronal membranes, stopping nociceptive activity potentials. While capsaicin acts as an agonist for the transient receptor potential vanilloid subtype 1 (TRPV1) channel at the level of nociceptors. Their adverse effects generally consist of localized erythema and skin rash [108].

According to the FDA, patches with 5% lidocaine are classified in category B regarding the potential risk following administration to pregnant women. There are no studies on the human species that affirm the safety of the drug, but in studies on rats, the transdermal application of lidocaine in a dose of 30 mg/kg did not produce adverse reactions in the fetus. In the Collaborative Perinatal Project, in the 293 children born to mothers exposed to lidocaine in the first trimester of pregnancy, no increase in the frequency of congenital anomalies was observed. The safety of topical capsaicin during pregnancy or breastfeeding has not been established in humans. Animal studies have not shown fetal teratogenicity in the development of the fetus, and knowing that small amounts are absorbed through the skin, it is not considered to cause adverse reactions in humans [109].

## 4. Treatment of Musculoskeletal Pain by Alternative and Complementary Methods

Exercise and rest are the basis for treating chronic musculoskeletal pain. The main goals of physical activity are to regain strength, endurance, flexibility and, especially, to reduce chronic pain by modulating the biochemical processes in the body [110]. The side effects of physical therapy are rare and are limited to musculoskeletal injuries, increased pain, dehydration, hypo or hyperthermia and, in more severe cases, respiratory and cardiac problems [111]. Physical therapy is not contraindicated for pregnant women. The benefits of this type of treatment in combating pain far outweigh the risks, being an easy way to improve quality of life. However, it is important to note that in the special condition of pregnancy, physical therapy is recommended to be performed only under the strict supervision of a person authorized in this field [48]. There are various physical therapy programs that differ in intensity, frequency, design and area of action. In order to relieve lower back pain, relaxation exercises and the correction of the sleeping position are recommended. Exercise in water is used by pregnant women to combat the pain caused by the mechanical pressure placed by the pregnant uterus on the muscles [112,113].

Non-invasive therapy is most often preferred over drug treatment, and surgery is not considered an option during pregnancy. Specialists in this field use several treatment techniques for pregnancy-associated lower back pain, including passive treatments, such as rest, but especially active treatments, such as exercise [40].

### 4.1. Physical Exercises and Water Gymnastics

The effects of exercise during pregnancy and postpartum have been evaluated in several randomized controlled trials [114,115,116,117,118,119,120,121,122,123,124,125]. In most studies, exercise has been associated with other methods of physiotherapy, such as manual therapy, pelvic girdle strengthening and physical therapy. In most cases where these physiotherapy techniques have been used, positive effects on pain in pregnancy have been reported compared to control groups or those undergoing only standard care [114,116,122,125]. However, one study found that the group receiving acupuncture treatment had lower pain intensity and increased physical activity than standard physical therapy treatment, home exercise, pelvic girdle exercises, and massage [115]. Exercise-only therapy is an insufficient method of relieving or combating pain in pregnancy, but it is still more successful than not intervening at all [126]. No significant side effects of the stabilization exercises were reported, with the exception of three women who reported a high intensity of pain during the exercises [127]. Some clinical trials conducted in this regard have highlighted preventive exercises. In a randomized clinical trial conducted in two groups, it was observed that pregnant women in the group who exercised for 12 weeks (aerobic and strengthening exercises) both at home and in joint sessions requested a lower proportion of medical leave due to pelvic pain than women in the group that received only standard care associated with adequate information from a specialist [119]. Furthermore, another study found that exercise decreased the incidence of lower back pain at the end of pregnancy, with increased physical function [128]. Another study compared lumbopelvin pain experienced by pregnant women exercising in water and on land. It was observed that after performing the exercises in the water, the pain was less intense and less medical leave was needed compared to the group who performed exercises on land [129].

In another study, water gymnastics proved effective. Specifically, performing exercises in the water once a week in the second half of pregnancy has significantly reduced back and back pain, with no risk of urinary or vaginal infections following this therapy. Therefore, the reduction of back pain has also reduced the need for medical leave because of this [111].

### 4.2. Manual Therapy

Massage therapy and chiropractic care are effective treatment options for pregnant women suffering from lower back and pelvic pain. Regarding side effects, very few were reported, not affecting the baby, lumbar spine or pelvis [130,131].

With the development of pregnancy, especially in the third trimester, the center of gravity of the pregnant woman moves to the anterior and there is an increase in lumbar lordosis, which has the effect of hyperactivity of the pelvic muscles and the hypermobility of the joints. In this case, a chiropractor can intervene to help the pregnant woman reduce the mechanical pain, and even giving advice about special pillows can significantly reduce these problems in the pregnant woman. In addition, high pressure is exerted on the lower limbs, requiring proper footwear, orthoses or massage to provide the necessary comfort [132]. Massage therapy has been shown to be effective in combating chronic and subacute back pain in the general population [133]. Evidence for the effectiveness of manual therapy in relieving pain in pregnant women is limited. A randomized clinical trial investigated craniosacral therapy. It was reported that this therapy in combination with standard therapy reduced the intensity of pain in the morning, with a slight impairment of function compared to standard therapy alone [134]. Another study evaluated manual therapy, and osteopathic therapy showed a decrease in pain intensity and disability compared to general therapy without or with sham-ultrasound [135]. Osteopathic therapy approaches the whole body, using classic techniques of light pressure, stretching and resistance [136]. Pregnancy produces major changes in the musculoskeletal system, from the straining of ligaments and the decrease in range of motion to an increase in muscle tension, causing pain. The main aim of osteopathic treatment for pregnant women is to relieve pain and support general function. Treatment includes soft tissue massage and the targeted mobilization of affected muscles and joints. Gentle exercises at home are recommended to manage symptoms but also to prepare the mother for the birth and for the life that follows. In addition, osteopathic manipulative treatment is defined as a manually guided force therapy by an osteopathic physician to support homeostasis and improve physiological function that has been affected by somatic dysfunction [137].

### 4.3. Pelvic Belt

Another method of treatment for the removal of musculoskeletal pain in pregnant women is the use of the pelvic girdle.

Randomized clinical trials have shown that using this method in women with pelvic girdle pain reduced their disability and pain intensity compared to women who only exercised or received general treatment information [117].

Another randomized study in women with symphyseal pain showed that wearing a stiff belt decreased pain in all patients, as well as in the group of those who exercised, but the combination of the two methods did not show an improvement in reducing pain intensity [138]. Moreover, in another study, lumbopelvic pain was reduced with the help of two non-rigid belts, one of which showed greater efficacy, although there were no significant differences between the two [139].

### 4.4. Acupuncture

Acupuncture is another technique for treating pain. Studies in this direction have shown that needle therapy used in pregnant women has significantly reduced pain and increased function [114,115,140,141]. One study found that the effectiveness of acupuncture is similar to that of therapy by combining training, stretching for postpartum pain regression with manual therapy [142]. In addition, acupuncture did not cause serious adverse reactions in either mother or fetus [127,143]. The main treatment methods used to combat musculoskeletal pain are shown in Figure 6.

### 4.5. Electrotherapy

Exercise and manual therapy should be the main treatment methods for relieving pelvic and lower back pain during pregnancy. However, if these techniques are not enough or are limited due to pregnancy, electrotherapy can be used [144]. TENS (transcutaneous electrical nerve stimulation) is a non-pharmacological method that uses a low-intensity electrical current to inhibit the transmission of pain information to the central nervous system [145]. The effectiveness of TENS was evaluated in a randomized clinical trial. The results showed a decrease in pain intensity and an increase in function in pregnant women who used this therapy compared to groups who used exercise or who received paracetamol [146]. Evidence is limited, requiring more clinical trials.

Likewise, TENS can be used to reduce pain during labor, it has been shown to be a safe technique for both mother and newborn [147].

### 4.6. The Benefits of Physiotherapy

Exercise is recommended for pregnant women to limit weight gain but also to maintain cardiovascular function and to prevent chronic diseases. Exercises on land or in the water, as well as manual therapy, acupuncture and electrotherapy, have been shown to be beneficial in combating muscle pain during pregnancy, as presented. Furthermore, exercise has beneficial effects on other levels. Significant weight gain during pregnancy increases the risk of many pathologies, from gestational diabetes to pregnancy-induced hypertension. Thus, it has been found that frequent exercise during pregnancy reduces the risk of obesity and the association with a healthy diet is the best method of weight control [148]. Another positive finding of physiotherapy in pregnancy is the benefit regarding gestational diabetes. A study conducted by Barakat and his team found that moderate exercise improves maternal glucose tolerance without any signs of gestational diabetes [149]. Moreover, exercise in women with gestational diabetes has reduced the need for insulin [150]. During pregnancy, the woman undergoes various anatomical and physiological changes that can even lead to an increased rate of urinary stress incontinence. In this regard, there have been studies that have shown the benefits of pelvic floor exercises in preventing and treating this problem in pregnancy [151,152].

However, one aspect that should not be overlooked is the psychological impact of physiotherapy. Clinical studies have concluded that exercise is an ameliorating factor in pregnancy and postpartum depression, improving the quality of life and the perception of pain and general health [153,154,155].

## 5. Conclusions

Many women experience musculoskeletal problems during pregnancy, which often limit their daily activities, so therapeutic intervention is needed to prevent or combat pain. The information collected showed that alternative and complementary methods decrease the intensity of pain and in addition help to improve musculoskeletal function. Although there is clear evidence of the effectiveness of exercise therapy in combating lower back and pelvic pain in pregnancy, it is not possible to specify exactly what kind of exercises are suitable for each pregnant woman, the exercises being different in the studies. To be able to compare the effectiveness of existing therapies, it is necessary to select groups according to the same set of criteria. Further studies are needed to strengthen the effectiveness of physical activity, as well as more robust data on the effectiveness of electrotherapy and craniosacral and osteopathic therapy, as data are limited in this direction. An optimal level of intensity of physical activities must be established to maintain the safety of the woman and the fetus, and more detailed studies are required to determine the relationship between the type of exercise, its dose, intensity, and desired response of preventing or treating a problem.

## Figures and Tables

**Figure 1 medicina-58-01115-f001:**
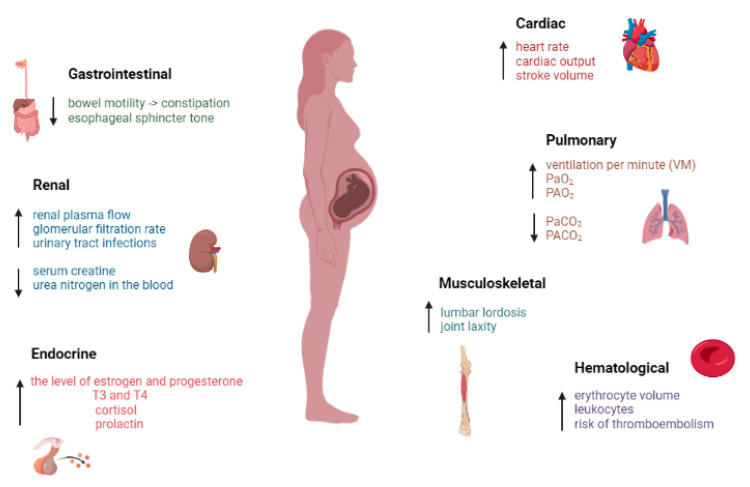
The main physiological and anatomical changes of pregnancy. (↑—increase, ↓—decrease).

**Figure 2 medicina-58-01115-f002:**
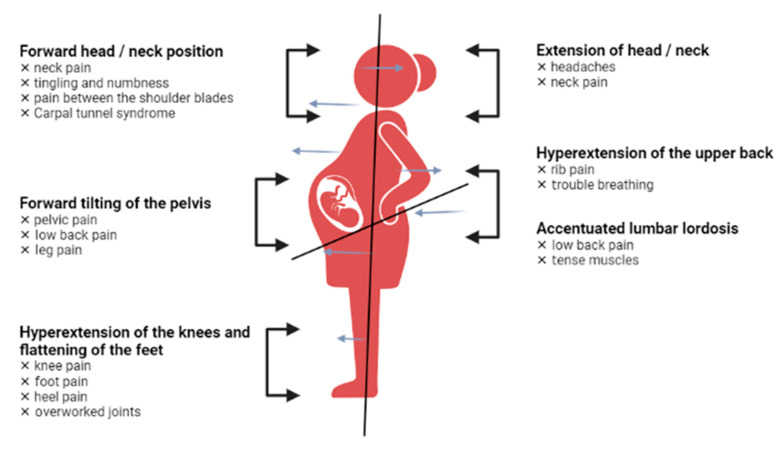
Postural changes in pregnant women.

**Figure 3 medicina-58-01115-f003:**
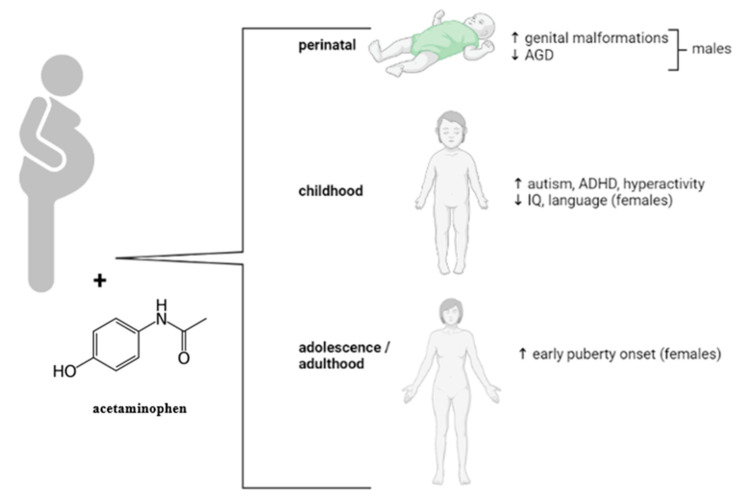
Acetaminophen exposure in potential pregnancy-associated effects observed in human studies. (↑—increase, ↓—decrease).

**Figure 4 medicina-58-01115-f004:**
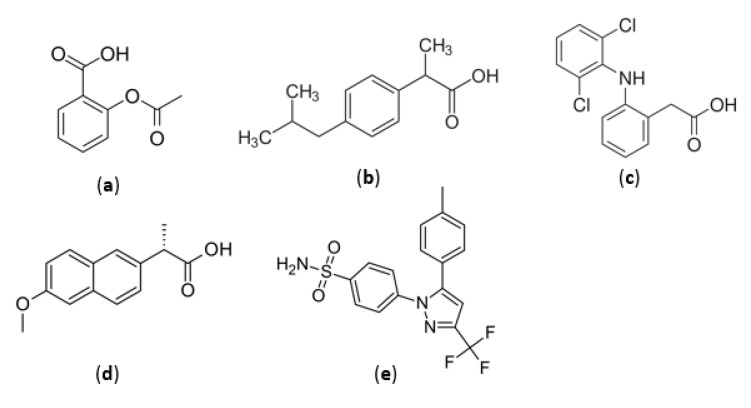
Chemical structures of non-steroidal anti-inflammatory agents: (**a**) aspirin, (**b**) ibuprofen, (**c**) diclofenac, (**d**) naproxen and (**e**) celecoxib.

**Figure 5 medicina-58-01115-f005:**
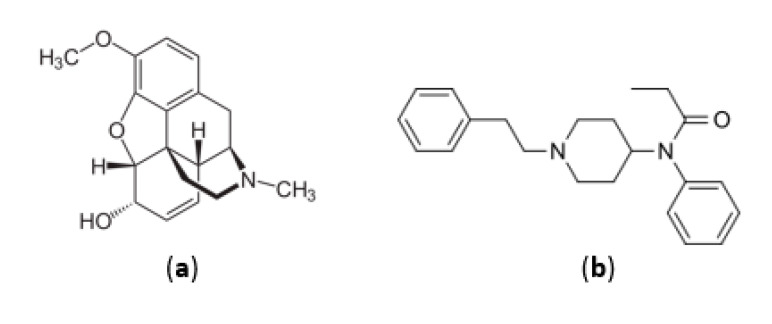
Chemical structure of opioids: (**a**) codeine and (**b**) fentanyl.

**Figure 6 medicina-58-01115-f006:**
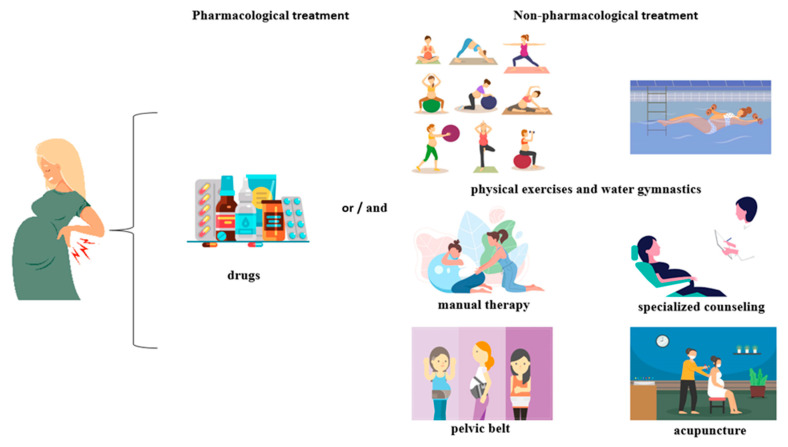
Pharmacological and non-pharmacological treatment of pain in pregnant women.

**Table 1 medicina-58-01115-t001:** The FDA’s classification of drugs used in pain management regarding pregnancy risk.

Classification	Definition	Examples	References
Category A	Controlled studies in pregnant women have not shown a risk to the fetus.	-	
Category B	Animal studies have not shown toxic effects on the fetus, but there are no controlled clinical trials in humans. Animal studies have shown toxic effects but have not been confirmed in studies in pregnant women.	Acetaminophen	[52,53,54,55]
		Oxycodone	
		Lidocaine	
		Dexamethasone	
Category C	Animal studies have shown teratogenic risk, but no controlled studies have been performed in pregnant women.	Piroxicam	[56,57,58,59,60,61,62,63,64,65]
		Codeine	
		Hydrocodone	
		Diclofenac	
		Ibuprofen	
		Naloxone	
		Ketoprofen	
		Celecoxib	
		Naproxen	
		Tramadol	
Category D	Positive evidence of fetal risk but their use is accepted, the benefit of the mother outweighs the risk of the fetus.	Acetyl salicylic acid	[66]
Category X	Studies in both animals and humans have shown fetal malformations or no evidence. The risks outweigh the benefits, they are contraindicated in pregnancy.	Ergot derivatives	[67]

## Data Availability

The data supporting the findings of the study are available within the article.

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
