# Peer review of "The Main Changes in Pregnancy—Therapeutic Approach to Musculoskeletal Pain"

_medicina, 2022, doi:10.3390/medicina58081115_

Round 1

Reviewer 1 Report

The aim of the  article, as well as the inclusion/ exclusion criteria must be detailed clearly 

I considered that the authors need to descibe more exhaustive the causes of pain, and the compatible manners of diagnosis .

For a review I considered more suitable to use score of pain/ fatigue/ functional scale and to describe the percentages of improvement regarding  different therapeutical approaches. ( more data from studies with numbers)

I will attached below the verified version of the manuscript with specific comments

Author Response

Thank you very much for the time allocated for the review of the manuscript “ The main changes in pregnancy - therapeutic approach to mus-culoskeletal pain” as well as for your pertinent observations.

Please see below a point-by-point response to your comments:

Reviewer #1:

Please describe clearer the objective of the article, as well as the inclusion/exclusion criteria.

Response: Thank you very much for your comment. We have adjusted it.

Please provide more data regarding the causes of the pain during pregnancy, eventual the methods of diagnose (Ultrasound/EMG/MRI?)

Response: Thank you very much for the comment. We have made the additions.

Please provide more information regarding sacroiliac bone marrow edema in pregnant female and the differential diagnosis with spondylarthritis

Response: We have taken the reviewer’s comment and we have adjusted it.

Please describe sacroiliac/disk hernia?

Response: Thank you very much for the suggestion. We have made the change.

Please describe more information regarding the good outcome of medication regarding the pregnant patient pain or functionality, not only the adverse effects.

Response: Thank you very much for your comment. The text contains information on the beneficial analgesic effect in pregnant women, but with more emphasis on adverse effects in order to be aware of the importance of knowing the risk/benefit ratio of pharmacological medication.

The negative influence of NSAIDs on fertility, please describe risk of unruptured of ovarian follicle regarding the females which use usually NSAIDs.

Response: Thank you very much for your comment. We have made the additions.

NSAIDs may increase the risk of permanent pulmonary hypertension of neonate and the subject must be developed.

Response: Thank you very much for the suggestion. We have changed it.

LDA during pregnancy and the favorable outcome regarding the preeclampsia.

Response: We have taken the reviewer’s comment and we have made the change.

Please describe the classes, not chemical structure and extrapolate the transplacental passage regarding each subclass.

Response: Thank you very much for the suggestion. We have added a class division of NSAIDs according to structure, but the text refers to the structures of the most used NSAIDs for pregnant women.

Nonprescription (over-the-counter) capsaicin patches or capsaicin cream?

Response: We have taken the reviewer’s comment and we have made the additions: “Transdermal therapies”.

Please provide more info regarding the type of exercises, please mention another study.

Response: Thank you very much for your comment. We have made the adjustments.

Data from studies? The frequency of procedure? The trimester of pregnancy?

Response: Thank you very much for your comment. These aspects are not presented in all the analyzed studies, in general the procedures (physical exercises on land, water exercises, aerobics, massage) are performed once or twice a week throughout the pregnancy.

Osteopathic therapy please add more details regarding the use during pregnancy.

Response: Thank you very much for the suggestion. We have made the additions.

We hope you find the revised manuscript acceptable for publication. Thank you once again for your consideration.

Best regards,

Alexandra Scurtu,

Timisoara,

27th of July, 2022

Reviewer 2 Report

Thank you for the opportunity to review this manuscript. 

Although the information presented is interesting, the manuscript does not comply with the journal's submission guidelines: "We do not have strict formatting requirements, but all manuscripts must contain the required sections: Author Information, Abstract, Keywords, Introduction, Materials & Methods, Results, Conclusions, Figures and Tables with Captions, Funding Information, Author Contributions, Conflict of Interest and other Ethics Statements. Check the Journal Instructions for Authors for more details". 

Perhaps this is the main problem of manuscript presentation. 

Scientific literature must be replicable. That is why it is important that the authors indicate, in an exhaustive way, the steps taken during their research. 

Depending on the type of study, the journal "Medicine" indicates that: "Reviews: These provide concise and precise updates on the latest progress made in a given area of research. Systematic reviews should follow the PRISMA guidelines".

Like the journal, I strongly recommend following these guidelines for the preparation and submission of manuscripts.

Author Response

Thank you very much for your comment. We have made the changes to the abstract according to the requirements. If you think that additions are still necessary, please let me know.

Round 2

Reviewer 1 Report

The article is really improved, all my recommendations were strongly  respected. I consider the actual version compatible with publication .

Reviewer 2 Report

Thank you for the opportunity to review this manuscript. 

Although the information presented is interesting, the manuscript does not comply with the journal's submission guidelines: "We do not have strict formatting requirements, but all manuscripts must contain the required sections: Author Information, Abstract, Keywords, Introduction, Materials & Methods, Results, Conclusions, Figures and Tables with Captions, Funding Information, Author Contributions, Conflict of Interest and other Ethics Statements. Check the Journal Instructions for Authors for more details". 

Perhaps this is the main problem of manuscript presentation. 

Scientific literature must be replicable. That is why it is important that the authors indicate, in an exhaustive way, the steps taken during their research. 

Depending on the type of study, the journal "Medicine" indicates that: "Reviews: These provide concise and precise updates on the latest progress made in a given area of research. Systematic reviews should follow the PRISMA guidelines".

Like the journal, I strongly recommend following these guidelines for the preparation and submission of manuscripts.